# Know your epidemic, know your response: Early perceptions of COVID-19 and self-reported social distancing in the United States

**Alberto Ciancio**[1,2]*, **Fabrice Kämpfen**[1], **Iliana V. Kohler**[1], **Daniel Bennett**[3], **Wändi Bruine de Bruin**[3], **Jill Darling**[3], **Arie Kapteyn**[3], **Jürgen Maurer**[2], **Hans-Peter Kohler**[1]*

1 Population Aging Research Center and Population Studies Center, University of Pennsylvania, Philadelphia, PA, United States of America, 2 Department of Economics, University of Lausanne, Vaud, Switzerland, 3 Center for Economic and Social Research, University of Southern California, Los Angeles, CA, United States of America

* ciancio@sas.upenn.edu (AC); hpkohler@pop.upenn.edu (HPK)

**Data Availability Statement:** The code and data are available at https://uasdata.usc.edu. Any interested reader needs to register for an account

## Abstract

As COVID-19 is rapidly unfolding in the United States, it is important to understand how individuals perceive the health and economic risks of the pandemic. In the absence of a readily available medical treatment, any strategy to contain the virus in the US will depend on the behavioral response of US residents. In this paper, we study individual's perceptions on COVID-19 and social distancing during the week of March 10–16, 2020, a week when COVID-19 was officially declared to be a pandemic by WHO and when new infections in the US were more than doubling every three days. Using a nationally representative sample of 5,414 respondents 18+ years of age from the Understanding America Study (UAS), we find that perceptions about COVID-19 health risks and economic consequences in the US population were largely pessimistic and highly variable by age and education. US residents who are young and do not have a college degree perceived a *lower* risk of getting infected but a higher probability of running out of money than others. Most individuals reported taking some steps to distance themselves from others but important differences emerge by gender and by source of information on COVID-19. Using state and day fixed-effect regressions, we show that perceptions of the health risks closely followed the number of COVID-19 cases in the country, and perceptions of the economic consequences and the prevalence of social distancing were driven upwards by both national and state-level cases. Unless addressed by effective health communication that reaches individuals across all social strata, variations in perceptions about COVID-19 epidemic raise concerns about the ability of the US to implement and sustain the widespread and restrictive policies that are required to curtail the pandemic.

## Introduction

"Know your epidemic, know your response", a rallying theme of HIV prevention with applicability to the COVID-19 pandemic, highlights the fact that individuals' knowledge about infection risks is critical for their ability to respond to the pandemic with appropriate risk reduction strategies. It also supports the restrictions on individuals' lives that are implemented to curtail

and submit a COVID-19 Proposal through the UAS website. Others would be able to access these data in the same manner as the authors. The authors did not have any special access privileges that others would not have.

**Funding:** We gratefully acknowledge the support of the Population Aging Research Center (PARC) and Population Studies Center (PSC) at the University of Pennsylvania, which are funded by the U.S. National Institutes of Health through grants NIA P30AG12836 and NICHD R24 HD044964 respectively to HPK. Data collection was funded and conducted by the University of Southern California's Center for Economic and Social Research to AK. Wändi Bruine de Bruin was supported by USC's Schaeffer Center for Health Policy and Economics, and the Swedish Foundation for the Humanities and the Social Sciences (Riksbankens Jubileumsfond) Program on Science and Proven Experience 'Science and Proven Experience' The funders had no role in study design, data collection and analysis, decision to publish, or preparation of the manuscript.

**Competing interests:** The authors have declared that no competing interests exist.

the spread of the virus [1–4]. Indeed, with a lack of effective medical treatments or a vaccine, the best hope to fight the COVID-19 pandemic is through changes in individual social behavior. The extent to which individuals are willing to implement social distancing crucially depends on their understanding of the risks associated with the epidemic. Individuals who have perceptions of high health risks associated with Covid-19 may be more likely to practice social distancing while those who have perceptions of big economic risks may be less in favor of restrictive measures. Observers now recognize that the United States was late to adopt proactive measures to address COVID-19 in the early stages of the epidemic [5]. This is problematic as individual behavior in the early phases of the epidemic seems to be critical to flatten the daily infection cases curve [6–9]. Early perceptions of the risks associated with COVID-19 are important as they determine behavior in these early crucial phases of the epidemics [10].

With growing evidence that lower income individuals are dying from COVID-19 at disproportionately high rates [11, 12], it is also particularly important to study the heterogeneity in perceptions and behaviors. While the gradient certainly depends on several factors like unequal access to jobs that allow working from home, different beliefs may reinforce or diminish the health gradient by socioeconomic status.

Using data from the COVID-19 focused questionnaire of the Understanding America Study [13] collected during March 10–16, 2020, our study is the first to use weighted nationally-representative data to document individuals' perceptions of the risks and consequences of the COVID-19 pandemic in the United States. In particular, the objectives of this study are to show the average levels of risk perceptions of getting infected with Covid-19, the mortality and economic risks associated with the pandemic as well as the variation in these risk perceptions by basic demographic characteristics. Similarly, the study presents average levels and heterogeneity in self-reported social distancing measures. Finally, the paper explores how changes in individual risk perceptions and behavioral response correlate with changes in COVID-19 confirmed cases over time at the country and state level.

## Data and methods

The Understanding America Study (UAS) is a nationally representative [10, 14, 15] probability-based Internet panel of approximately 8,800 respondents representing the entire United States population. Participants are recruited offline through random sampling of residential addresses and consequently invited to complete surveys online [16, 17]. Respondents without prior access to the Internet receive a tablet and broadband Internet [18]. As a probability-based internet sample, UAS is distinguished from convenience internet panels, where respondents are recruited only from among existing Internet users, with unknown inclusion probabilities. Several studies have shown that even with relatively low response rates probability-based Internet panels provide superior information to convenience panels [19, 20]. A comparison of UAS data with Current Population Survey (CPS) and Health and Retirement Study (HRS) data [21] shows that UAS data line up quite well with the CPS on a number of common variables, and there is no clear evidence that UAS compares less favorably to the CPS than the HRS, which is traditionally considered as a gold standard in social research.

Since its start in 2014, UAS has collected more than 250 online surveys on various topics, ranging from cognitive abilities, environment, consumer behavior and politics to name but a few, for which the data is publicly available. The online survey was approved by USC's Institutional Review Board, as part of the UAS, and informed consent was obtained from all participants. More information about the study and its design can be found at: https://uasdata.usc.edu/index.php.

Our analysis uses a sample of 5,414 respondents 18+ years of age who answered UAS survey #230 between March 10 and March 16, 2020. To be precise, on March 10, 8815 UAS respondents were invited to take the survey. By the end of the field period (March 31), 7145 respondents had answered the survey for a response rate of 81%. We focus on the early stages of the pandemic, that is the period from March 10 to 16, before all local or statewide stay-at-home orders were issued. Our analysis therefore excludes 1731 respondents who completed the survey between March 17 and March 31. In all analyses, we use post-stratification weights, generated through a raking algorithm, to align our study sample to the U.S. adult population aged 18 and older in terms of gender, age, race/ethnicity, education and geographic location (see https://uasdata.usc.edu/page/Weights [17]). S1 Table presents the weighted averages of the main demographic characteristics we use in our study. S2 Table in the Appendix shows that there are some statistically significant differences in the characteristics of individuals who completed the survey by March 16[th], and those who did not, although these differences are small in magnitude. In addition to usual sociodemographic characteristics, the survey also elicited from respondents subjective probabilities of COVID-19-related events—such as getting infected with coronavirus (SARS-CoV-2) or dying in case of infection—on a scale from 0 to 100, and whether respondents had taken steps to stay away from others such as avoiding public spaces and canceling work or social activities.

More specifically, the risk perceptions variables used in the analysis are derived from the following questions: 1. "*On a scale of 0 to 100 percent, what is the chance that you will get the coronavirus in the next three months*?"; 2. "*If you do get the coronavirus, what is the percent chance you will die from it*?"; 3. "*What is the percent chance that you will lose your job because of the coronavirus within the next three months*?"; 4. "*What is the percent chance you will run out of money because of the coronavirus in the next three months*?" Respondents answered these questions using a scale from 0 to 100. In the analysis, we rescaled the answers by dividing by 100 to interpret the variables as probabilities. Using responses to questions 1) and 2), we derive a measure of excess mortality due to Covid-19 by multiplying the chances of getting infected with the chances of dying from the virus conditioning of being infected. Note also that question 3 was asked only to respondents who had a job at the time of the interview (N = 3252).

We define a dichotomous measure of social distancing by exploiting the answers from the question "*Which of the following have you done in the last seven days to keep yourself safe from coronavirus in addition to what you normally do*?" We consider that a respondent was refraining from at least one social activity if the person reported to have done at least one of the following: "Canceled or postponed travel for work", "Canceled or postponed travel for pleasure", "Canceled or postponed work or school activities", "Canceled or postponed personal or social activities", "canceled a doctor's appointment", "avoided contact with people who could be high-risk", "avoided public spaces, gatherings, or crowds", "avoided eating at restaurants" or "worked or studied at home". Cronbach's alpha for the nine items that compose our social distance index is 0.83, which suggests that the items reliably measure the same construct–social distancing [22]. Moreover, single-item measures tend to have good predictive validity, as compared to multi-item measures [15, 23].

We complement UAS survey data with data on daily COVID-19 cases and deaths detailed at the state and national level that we obtained from John Hopkins University [24]. By linking UAS to this information, we can assess whether public information about the course of the pandemic in the place of residence impacts perceptions and social distancing. To this end, we regress each of our outcomes on the (log) cumulative number of COVID-19 cases and the cumulative number of deaths on the day of the interview. We run separate regressions for COVID-19 cases and deaths at the national and at the state level. We control for demographic characteristics and include state of residence fixed effects. In the regressions with cases and

deaths at the state level, we also include day of interview fixed effects. The STROBE guidelines were used to ensure the conformity of the reporting of our observational study [25].

## Results

Around mid-March, when the COVID-19 epidemic rapidly expanded in the United States, U. S. residents perceived on average a 20% chance of getting infected with the coronavirus during the next three months (Fig 1A and Table 1). These 3-months average infection risks of about one out of five persons is consistent with disease scenarios in which the containment of the disease is no longer possible and a large fraction of the U.S. population will ultimately get infected with the virus, while social distancing is able to flatten the curve and spread infections across an extended time period [26]. Yet, underneath this 20% average is considerable heterogeneity: 21% of U.S. residents perceive zero risk of getting infected in the next 3 months, while 21.8% perceive a 0.1–10% risk, 17.6% a 10.1–20% risk, and 39.7% a 20% or higher chance. Only a small fraction, 2.4%, perceives a 3-month infection probability of 80% or higher (Table 1). Neither the average infection risk, nor the fractions perceiving fairly low or fairly high infection risks differ substantially between the three states that had already widespread community transmission during the week of March 10–16 (Washington, California, New York) and the rest of the United States (Table 2). There are also no significant gender differences (Fig 2A). Important differences, however, emerge across education categories and age groups, with more educated and younger persons perceiving higher infection risks (Figs 1A and 2A). For example, U.S. residents in their 30s with a Bachelor's degree or higher perceive a 50% higher chance of being infected than their lower educated peers. There are also differences by individuals' sources of news: U.S. residents who use Fox News to obtain information about the COVID-19 pandemic report a 5 percentage points lower probability of being infected than U. S. residents who use CNN (Table 2). These differences persist controlling for basic demographic characteristics, state of residence and day of interview (S3 Table).

Individuals in the U.S. on average perceived a 14% chance of dying if infected with the coronavirus (Fig 1B and Table 1), and this average perceived mortality rate exceeds tenfold common estimates of the mortality risk due to COVID-19, while it is close to rates of death conditional on testing positive in areas where the health system was already failing and testing was restricted to the most ill patients during the pandemic peak (e.g., Lombardy in Italy). While the perceived mortality risk among U.S. residents from COVID-19 is relatively high, there is substantial variation across individuals: 23.6% perceive the chance of death if infected with coronavirus to be zero while 39.6% perceive this risk as 0.1–10%, 13.2% as 10.1–20% and 23.6% as 20% or higher (Table 1). These different perceptions of mortality if infected are importantly related to education and age. For example, individuals with a Bachelor's degree or higher estimate their chances of dying to be about 8 percentage points lower across primary adult ages than those with less education (Fig 1B). This education difference diminishes among older persons (Fig 1B), while they (accurately) perceive substantially greater mortality risks (Fig 2B).

Combining the risk of infection with the risk of death conditional on infection provides individuals' perceived excess mortality as a result of COVID-19. On average, individuals in the U.S. perceive a 3.8% chance of dying as a result of COVID-19 in the next three months (Fig 1C and Table 1), representing a significant perception of excess short-term mortality. These perceptions vary widely, with 31.6% expecting zero excess mortality from COVID-19, and 12% expecting an excess mortality in the next three months of 10% or more (Table 1). Perceptions of the excess mortality from COVID-19 are lower for more educated persons and higher for older individuals (Fig 1C), while there are no strong differences by gender (Fig 2C).

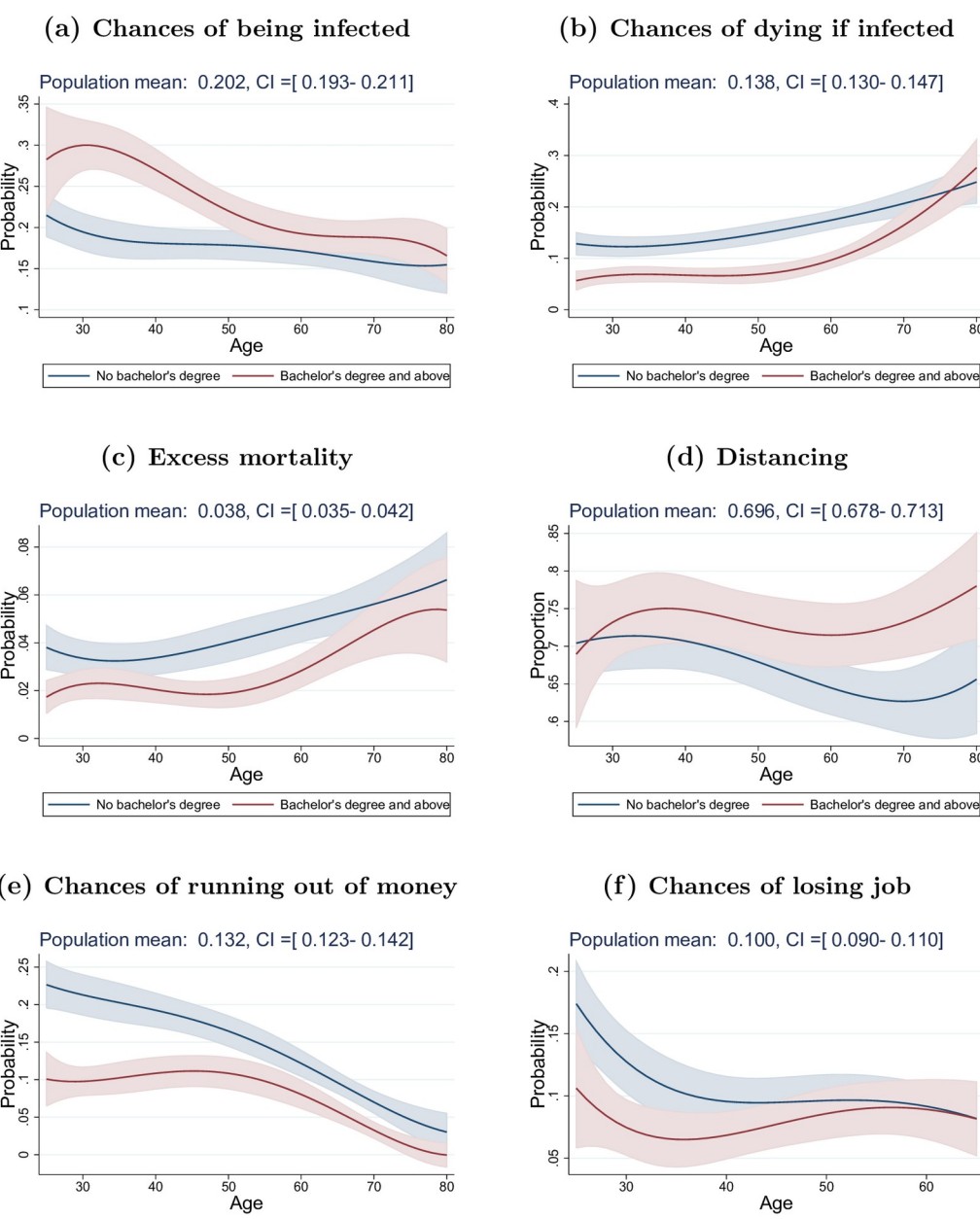

**Fig 1. Differences in perceptions and social distancing practice by education level.** Data coming from "Understanding America Study" (UAS) collected between March 10 and March 16, 2020. The plots show the marginal effects of an increase in age (1 year) on a) the chances of getting the virus within three months (top left), b) the chances of dying from the virus if infected (top right), c) excess mortality (middle left), d) whether individuals refrain from at least one social activity (middle right), e) the chances of running out of money because of the virus within three months (bottom left) and f) the chances of losing job within three months (bottom right), Effects for those with a Bachelor's degree and above are represented by red lines and by blue ones for others. Marginal effects for individuals aged between 25 and 80 are the results of weighted regressions that include quartic polynomial in age. Shaded areas represent 95% confidence intervals (robust standard errors).

There are important temporal patterns in the perceptions of COVID-19 during the week of March 10–16, 2020 when the identified cases increased from 959 to 4,632, and the reported COVID-19 deaths increased from 28 to 85 [24]. Importantly, our analyses of COVID-19

**Table 1. Summary statistics of the continuous measures used in the analysis (Sample size = 5,279).**

|  | Get Covid | Die from Covid | Excess Mortality | Lose Job | Out of Money |
|---|---|---|---|---|---|
| Population mean | .202 | .138 | .038 | .100 | .132 |
| Population median Interquantile range | .100 [.010-.320] | .038 [.002-.150] | .003 [0-.025] | 0 [0-.100] | 0 [0-.149] |
| *Weighted frequencies* | | | | | |
| Probability of 0 | .210 | .236 | .316 | .526 | .513 |
| Probability > 0 and < 10 | .218 | .396 | .567 | .199 | .172 |
| Probability > 10 and < 20 | .176 | .132 | .051 | .095 | .082 |
| Probability > 20 | .397 | .236 | .067 | .179 | .233 |
| Probability > 80 | .024 | .029 | .001 | .018 | .038 |

The table shows the weighted means and medians of the five continuous measures we use in the analysis (top panel): 1) the chances of getting the virus within three months, 2) the chances of dying from the virus if infected, 3) excess mortality, 4) the chances of losing job within three months and 5) the chances of running out of money because of the virus within three months. The bottom panel presents some characteristics of the weighted distribution of these measures. Data are from "Understanding America Study" (UAS) collected between March 10 and March 16, 2020.

perceptions during March 10–16 show that the rapid surge in caseload at the country level increased the perceived chance of getting infected with coronavirus: a one-unit increase in the log number of COVID-19 cases in the U.S. leads on average to an increase of about 0.055 in the perceived chances of getting infected by the virus (Table 3, top panel), which corresponds to a 3.8 percentage points increase for each doubling of the national U.S. COVID-19 caseload ($0.055 \times \ln(2)$). Neither the increase in national COVID-19 cases, nor the increase in deaths, was positively associated with expectations about dying conditional on being infected with COVID-19 or the perceived excess mortality as a result of COVID-19 (Table 3 and S4 Table). If anything, the increase in national cases had a negative (but imprecisely estimated) impact on the perceived probability of dying conditional on infection. This change in perceptions is consistent with a scenario where expanded testing across the country increasingly identifies milder cases with lower mortality risks conditional on infection.

As of March 10–16, U.S. residents had increasingly adopted social distancing measures to delay the spread of the virus: about 70% of people in the U.S. report to have taken steps to stay away from other persons ("social distancing") (Fig 1D and Table 1), and despite the widely-circulated images of young persons agglomerating on beaches and in bars, overall younger U.S. residents were at least as likely (based on self-reports) to have taken social distancing measures

**Table 2. Summary statistics (Sample size = 5,279).**

| CA, NY, WA | 95% conf. interval | | Other States | 95% conf. interval | | Used Fox News | 95% conf. interval | | Used CNN | 95% conf. interval | |
|---|---|---|---|---|---|---|---|---|---|---|---|
| 1) Get Covid | **.208** | .191 | .225 | **.200** | .189 | .210 | **.174** | .159 | .189 | **.228** | .212 | .244 |
| 2) Die from Covid | **.133** | .119 | .147 | **.140** | .130 | .150 | **.146** | .132 | .160 | **.143** | .129 | .157 |
| 3) Excess Mortality | **.036** | .030 | .042 | **.039** | .035 | .043 | **.036** | .030 | .041 | **.045** | .039 | .052 |
| 4) Lose Job | **.124** | .106 | .142 | **.093** | .081 | .104 | **.099** | .084 | .114 | **.105** | .090 | .119 |
| 5) Out of Money | **.164** | .146 | .182 | **.123** | .112 | .134 | **.139** | .123 | .155 | **.144** | .127 | .161 |
| 6) Distancing | **.776** | .747 | .805 | **.672** | .651 | .693 | **.684** | .653 | .715 | **.783** | .756 | .809 |

The table shows the mean and 95% confidence intervals by state levels of infection and by source of information of 1) the chances of getting the virus within three months, 2) the chances of dying from the virus if infected, 3) excess mortality, 4) the chances of losing job within three months, 5) the chances of running out of money because of the virus within three months and 6) whether individuals refrain from at least one social activity. High-infected states include California (CA), New York (NY) and Washington (WA). We use sample weights to make the survey representative of the U.S. population aged 18 and older. Data come from "Understanding America Study" (UAS) collected between March 10 and March 16, 2020.

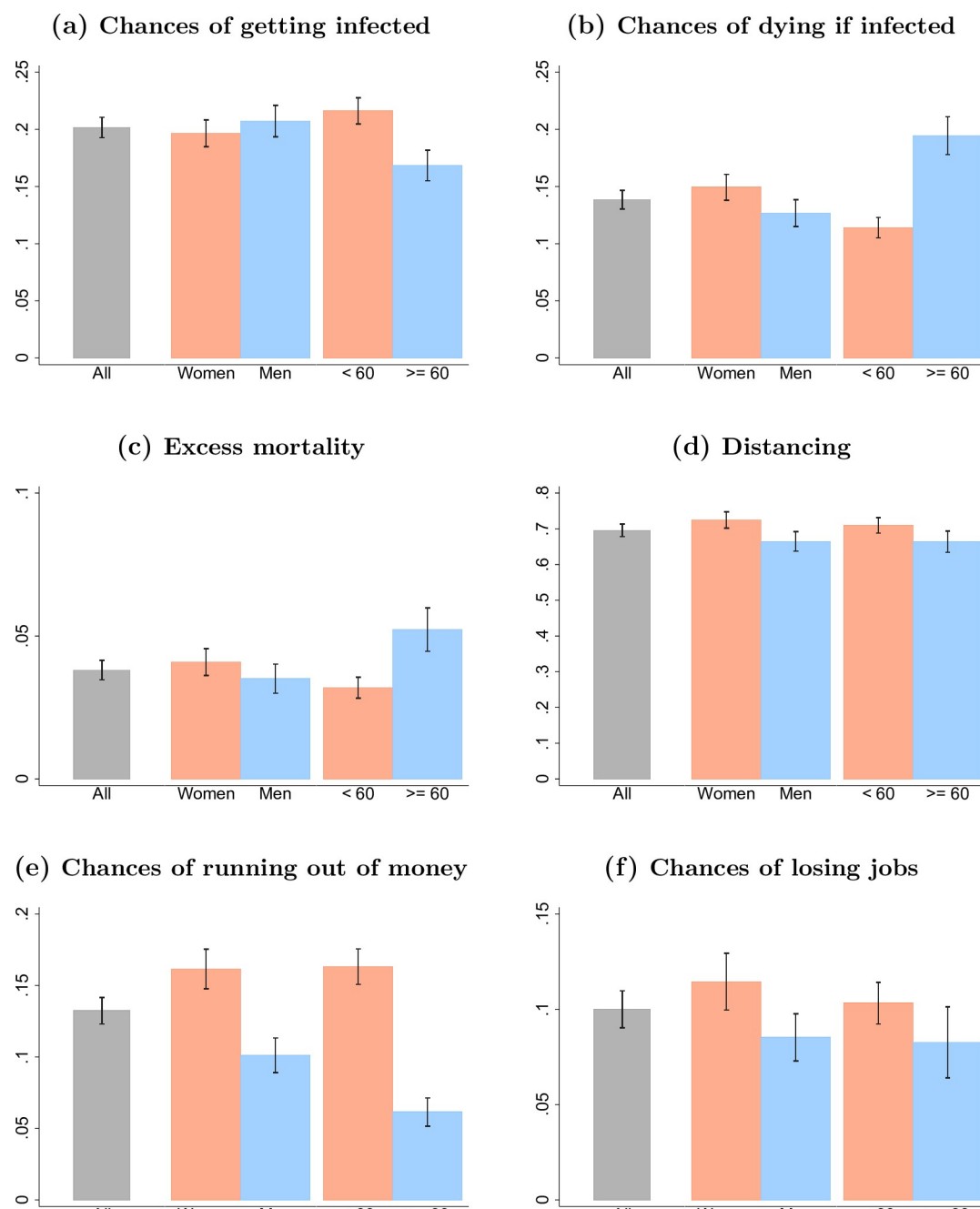

**Fig 2. Difference in perceptions and social distancing practice by age and gender.** Data come from "Understanding America Study" (UAS) collected between March 10 and March 16, 2020. Bar plots show weighted means for the six variables we consider in our analysis along with their confidence intervals, derived based on weighted standard errors. We present these statistics for all U.S. residents, and by gender and age categories (below or above 60).

as older persons (24). Women are more likely to engage in social distancing (Fig 2D), while educational differences only emerge among older people where college educated individuals were significantly more likely to practice social distancing than non-college educated individuals (Fig 1D). Using CNN as a source of information on coronavirus is associated with a 10 percentage points higher chance of engaging in social distancing than using Fox News (Table 2).

Individuals in the U.S. perceived an average 10% chance of losing their job within three months, and a 13% chance to run out of money in the next three months as a result of COVID-19 (Fig 1E, 1F and Table 1). These perceptions highlight a clear awareness of the short-term negative economic consequences as a result of the pandemic. There are again clear differences in these perceptions across education levels, with individuals holding at least a Bachelor's degree having about twice lower chances to run out of money at younger ages as a consequence of the virus outbreak compared to others (Fig 1E and 1F). Women are more concerned about these economic consequences than men (Fig 2E and 2F).

Importantly, U.S. residents' perceptions of the socioeconomic severity of the pandemic changed fundamentally during the week of March 10–16. During these seven days, the increase in cases at the national level induced people in the U.S. to revise upward the perceived likelihood of losing their job or running out of money as a result of COVID-19, and taking steps to stay away from others (Table 3). Our analyses show that each doubling of the U.S. caseload implied an increase of 4.2 percentage points increase in the chance of running out of money within three months, and a 14.3 percentage points increase in the likelihood that the respondents have taken steps to stay away from others. U.S. residents thus increasingly realized within merely seven days that COVID-19 might entail drastic socioeconomic consequences, such as losing one's job or running out of money within three months. These perceptions of the socioeconomic consequences of the epidemic, and the adaptation of social distancing measures, were particularly pronounced in states with the most severe epidemic (Table 3, bottom panel). Specifically, the increase in caseload in a respondent's state of residence accelerated worries about running out of money (2.4 percentage points for each doubling of cases) and increased social distancing (3.7 percentage points for each doubling of cases) above and beyond the increase in cases at the national level. The results persist in analyses that control for whether local authorities have implemented measures requiring social distancing.

**Table 3. Effects of Covid cumulative cases on Covid risk perceptions and social distancing.**

| | (1) | (2) | (3) | (4) | (5) | (6) |
|---|---|---|---|---|---|---|
| **Get Covid** | | **Die from Covid Excess** | **Mortality** | **Lose Job** | **Out of Money** | **Distancing** |
| | | | Regressions with state fixed effects | | | |
| US cases (log) | 0.055*** | -0.017* | 0.001 | 0.054*** | 0.060*** | 0.206*** |
| | [0.029,0.081] | [-0.036,0.002] | [-0.007,0.009] | [0.027,0.082] | [0.031,0.089] | [0.160,0.252] |
| | | | | | | |
| Regressions with state and day fixed effects | | | | | | |
| State cases (log) | 0.015 | -0.001 | 0.001 | 0.021 | 0.034** | 0.054*** |
| | [-0.017,0.047] | [-0.021,0.018] | [-0.011,0.013] | [-0.010,0.051] | [0.001,0.067] | [0.014,0.094] |
| Observations | 5272 | 5270 | 5269 | 3245 | 5301 | 5287 |

The table shows the effects of the log of the number of US and state of residence Covid-19 cases on 1) the chances of getting the virus within three months, 2) the chances of dying from the virus if infected, 3) excess mortality, 4) the chances of losing job within three months, 5) the chances of running out of money because of the virus within three months and 6) whether individuals refrain from at least one social activity. The top panel shows regressions that include state fixed effects while the bottom panel shows regressions that include both state and day of interview fixed effects. State cases are transformed using the inverse hyperbolic sine $log(x + (1 + x2)$^.5) which is similar to log but allows for zeros. Additional controls include age, gender and four categories of education. We use sample weights to make the survey representative of the U.S. population aged 18 and older. 95% confidence intervals in squared brackets are calculated using standard errors clustered at the state level.

* $p < 0.1$

** $p < 0.05$

*** $p < 0.01$. Data on perceptions and social distancing come from "Understanding America Study" (UAS) collected between March 10 and March 16, 2020. Data on the number of Covid-19 cases come from the Johns Hopkins University Center for Systems Science and Engineering.

## Limitations

The study has several limitations. First, our measure of social distancing is based on items that are self-reported and we have no possibility to verify the truthfulness of respondents' responses [10]. Second, we defined someone as practicing social distancing if the person refrained from doing at least one of the social activities listed in the survey. Obviously, there are other ways to derive a measure of social distancing from these various social activity items. Still, in the early stages of the pandemic and before any stay-at-home order was issued, a significant share of US residents was still only marginally impacted by mobility restrictions and had not taken yet meaningful steps to adapt their social life. In fact, we observe important variation in our social distancing measure with around 30% of respondents who did not refrain from doing any of the social activities that compose the index. Finally, even if states stay-at-home orders were all enacted after our period of study, we acknowledge that the effect of the number of COVID-19 cases on perceptions and social distancing was partially mediated by governments actions, such as school closures and state emergency declarations, which encouraged individuals to adopt social distancing measures particularly in areas with a rapidly growing number of cases. An evaluation on how these policies affects risk perceptions and social distancing is beyond the scope of this paper. It is also worth to notice that knowledge about the pandemic and related behavior has changed a lot from March 2020 and that we should not extrapolate our findings to later stages of the pandemic in the U.S.

## Discussion

We document that during the week of March 10–16, the COVID-19 pandemic fundamentally affected the perceptions of U.S. residents about the health risks and socioeconomic consequences of the pandemic. During this week, it seems, "*everything changed*." Not only did the pandemic spread rapidly across the United States, but U.S. residents started to realize that the threat was real. Our key results are that in early to mid-March 2020: 1. State fixed effects regressions show that increasing COVID-19 caseloads heightened perceptions of infection risks and excess mortality risks, concerns about the economic implications increased substantially, and behavioral responses became widespread as the pandemic expanded rapidly in the U.S.; 2. average perceptions about the coronavirus infection risks are broadly consistent with projections about the pandemic, while average expectations about dying conditional on infection and expectations about COVID-19-related excess mortality during the next months were possibly too pessimistic; 3. there is a high level of heterogeneity in risk perceptions: U.S. residents with no college education and those who use Fox News as main source of information had a substantially lower perception of the risk of getting infected with COVID-19.

The actual excess mortality in the U.S. as a result of COVID-19 remains very uncertain at this point [27–29] and will depend strongly on the success of social distancing and other disease containment efforts [26, 30]. Nevertheless, our results suggest that individuals in the U.S. are probably too pessimistic in that regard: while people in the U.S. had average expectations about getting infected within three months that are broadly consistent with the projected infected population in intermediate scenarios of how the pandemic would unfold, on average they overestimate the mortality conditional on getting infected. This pessimism was perhaps influenced by the high level of mortality experienced in other parts of the world such as in Northern Italy.

The high degree of heterogeneity in COVID-19 perceptions is disconcerting from the perspective of implementing and sustaining an effective societal response to the pandemic. The education gradient in expected infection risks entails the possibility of having different perceptions of the reality of the pandemic between people with and without a college education,

potentially resulting in two different levels of behavioral and policy-responses across individuals and regions. Unless addressed by effective health communication that reaches individuals across all social strata, some of the misperceptions about COVID-19 epidemic raise concerns about the ability of the United States to implement and sustain the widespread and harsh policies that are required to curtail the pandemic.

## Supporting information

**S1 Table. Demographics characteristics of the sample.**
(PDF)

**S2 Table. Compare the characteristics of respondents and non-respondents.**
(PDF)

**S3 Table. Heterogeneity by source of information.**
(PDF)

**S4 Table. Effects of COVID-19 cumulative deaths on COVID-19 risk perceptions and self-reported social distancing.**
(PDF)

## Author Contributions

**Conceptualization:** Alberto Ciancio, Fabrice Kämpfen, Iliana V. Kohler, Hans-Peter Kohler.

**Data curation:** Daniel Bennett, Wändi Bruine de Bruin, Jill Darling, Arie Kapteyn.

**Formal analysis:** Alberto Ciancio, Fabrice Kämpfen.

**Writing – original draft:** Alberto Ciancio, Fabrice Kämpfen, Iliana V. Kohler, Hans-Peter Kohler.

**Writing – review & editing:** Alberto Ciancio, Fabrice Kämpfen, Iliana V. Kohler, Daniel Bennett, Wändi Bruine de Bruin, Jill Darling, Arie Kapteyn, Jürgen Maurer, Hans-Peter Kohler.

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
