## [Decision Letter · Decision Letter 0]

18 Jun 2020

PONE-D-20-12921

Know Your Epidemic, Know Your Response: Early Perceptions of COVID-19 in the United States

PLOS ONE

Dear Dr. Ciancio,

Thank you very much for submitting your manuscript "Know Your Epidemic, Know Your Response: Early Perceptions of COVID-19 in the United States" (#PONE-D-20-12921) for review by PLOS ONE. As with all papers submitted to the journal, your manuscript was fully evaluated by academic editor (myself) and by independent peer reviewers. The reviewers appreciated the attention to an important health topic, but they raised substantial concerns about the paper that must be addressed before this manuscript can be accurately assessed for meeting the PLOS ONE criteria. Therefore, if you feel these issues can be adequately addressed, we invite you to submit a revised version of the manuscript that addresses the points raised during the review process. We can’t, of course, promise publication at that time.

We look forward to receiving your revised manuscript.

Kind regards,

Abdallah M. Samy, PhD

Academic Editor

PLOS ONE

**Journal Requirements:**

**Reviewers' comments:**

Reviewer's Responses to Questions

**Comments to the Author**

1. Is the manuscript technically sound, and do the data support the conclusions?

Reviewer #1: Partly

Reviewer #2: Partly

Reviewer #3: Yes

Reviewer #4: Partly

2. Has the statistical analysis been performed appropriately and rigorously? 

Reviewer #1: I Don't Know

Reviewer #2: Yes

Reviewer #3: Yes

Reviewer #4: Yes

3. Have the authors made all data underlying the findings in their manuscript fully available?

Reviewer #1: Yes

Reviewer #2: Yes

Reviewer #3: Yes

Reviewer #4: No

4. Is the manuscript presented in an intelligible fashion and written in standard English?

Reviewer #1: No

Reviewer #2: Yes

Reviewer #3: Yes

Reviewer #4: Yes

5. Review Comments to the Author

Reviewer #1: The study seems to be very interesting. Appreciate the efforts taken by the researchers for a timely and pertinent COVID related article.

The title of the study is catchy however it is not reflecting the variables properly thereby loosing the essence of the paper.

The study does mention about ethical committee permission and participant consent.

There is less clarity been given in the paper regarding variables under study.

The questionnaire its items were not fully comprehensive to reach a conclusive evidence.

Validity and reliability of the questionnaire is not mentioned in the paper.

There is occasional typos and grammatical errors.

The figures were ambiguous.

While assessing the perceptions it would be better if there is adequate number of items to investigate a behavioral response ;risk perception variable were assessed by using only a 4 item scale which lacks comprehensiveness of the information.

Sample size should be mentioned in the top of each table.

Reviewer #2: This is an interesting paper and the topic is worthy of further discussion.

My doubts are how the authors have called the data representative in the manuscript. They have given a comparison in the methodology as well which can be improved from a reader's perspective. Secondly the period of data collection was during the early stages of pandemic spread in the country and the same should be discussed while discussing the findings of the paper. The knowledge about pandemic and related behavior is bound to be affected by a number of confounders: baseline socio-economic parameters, financial difficulties during pandemic and the effect of social isolation. All these are dynamic and should be stated in the discussion section to enhance the context of study findings.

Reviewer #3: This is a very interesting and well meaning piece of work, conducted in very scientific manner.

My comments are as under:

1. Abstract- You have not described the study methodology in the abstract; Kindly consider briefly describing the same in the abstract, especially when PLOS has flexibility with its word limit

2. Its hard to distinguish between the degree of social distancing which was due to behaviour change and self imposed and that due to stringent local governmental regulations. If the same can be segregated, I feel the same would strengthen the article, else that can go as study limitation.

3. In the similar lines, I suggested that you may consider correlation of the data on social distancing with stringency of local regulations on movement and other regulatory measures.

4. I feel that you essentially need to improve the figure 1, which is quite blurred and uninterpretable

Wish you all the best.

Reviewer #4: This is an interesting study, however there are sveral major concersn with it

1. Authors should be carefully classify the social distancing. If they have considered participant is following social distancing if participant answered “yes” for any one of the option, it may lead to overestimation. Many work related travels would have cancelled due to lockdown and restrictions on travel around the world. Participants would have cancelled work related travel due restrictive measures taken by governing body, but may not be following any of social distancing measures. This may lead to bias in the study.

2. Without reading the methodology and results, it is difficult to understand Table 1. Complete the categories name in Table 1.

3. Figure 1 is not clear and could not make out the legends, which make it difficult to comment.

4. The authors need to adhere to STROBE statement for reporting observational studies.

5. Introduction: At the end of introduction section, it is important to explicitly mention the objectives of the study.

6. Materials and Methods: Probability based sampling and providing tablet and internet to the respondents who do not have them adds crucial value to the study and reduces selection bias.

The Understanding America Study has nearly 8500 participants. The authors have used data from 5414. Explanation should be given as to how they were chosen. If 5414 were the number of respondents for this survey, then characteristics of non-respondents should be compared to respondents.

7. Results:

i. The first line in the results reads as follows: “U.S. residents perceived on average a 20% chance of getting infected with the coronavirus during the next three months”. The table shows that there is considerable difference between mean (20%) and median (10%). It would be better to give SD and IQR to understand the distribution, to better interpret the results.

ii. The authors have discussed some of the results in this section of the manuscript. These needs to be included in the discussion section.

8. Discussion:

i. The discussion section needs to be in detail and include a summary the key results, explanation for the results that have been obtained.

ii. The number of cases and deaths in the one-week period during which the study was conducted rose rapidly. The authors need to elaborate how this could have affected perception of risk of infection and death.

iii. The authors explain the excess perception of mortality risk to pessimism among the people… Could this be because of the situation in Italy at that time.... A question examining how much of the perception of the individuals is related to the situation experienced by other countries must be included...

iv. The authors have not included strengths and limitations of the study

v. The authors have failed to explain how perception of the people would sway the response to COVID-19.

6. PLOS authors have the option to publish the peer review history of their article (what does this mean?). If published, this will include your full peer review and any attached files.

Reviewer #1: No

Reviewer #2: Yes: Dr. Mohit Varshney

Reviewer #3: Yes: Dr. Neeraj Raizada

Reviewer #4: No

---

## [Author Response · Author response to Decision Letter 0]

1 Aug 2020

Dear editor and reviewers, we attached a letter named "Response to reviewers" where we provide a detailed answer to your comments.

For the figures, I used PACE as requested. However, the quality of the figures in this format is not great. Please let me know if you have any suggestion about this. I can also provide the original pdf if you like.

---

## [Editor Report · Decision Letter 1]

17 Aug 2020

Know Your Epidemic, Know Your Response: Early Perceptions of COVID-19 and Self-Reported Social Distancing in the United States

PONE-D-20-12921R1

Dear Dr. Ciancio,

We’re pleased to inform you that your manuscript has been judged scientifically suitable for publication and will be formally accepted for publication once it meets all outstanding technical requirements.

Kind regards,

Abdallah M. Samy, PhD

Academic Editor

PLOS ONE

---

## [Editor Report · Acceptance letter]

28 Aug 2020

PONE-D-20-12921R1 

Know Your Epidemic, Know Your Response: Early Perceptions of COVID-19 and Self-Reported Social Distancing in the United States 

Dear Dr. Ciancio:

I'm pleased to inform you that your manuscript has been deemed suitable for publication in PLOS ONE. Congratulations! Your manuscript is now with our production department. 

Kind regards, 

on behalf of

Dr. Abdallah M. Samy 

Academic Editor

PLOS ONE